# Design Pattern Elicitation Framework for Proof of Integrity in Blockchain Applications

**Kawther Saeedi** [1,*], **Monirah Dakilallah Almalki** [1], **Dania Aljeaid** [1], **Anna Visvizi** [2] and **Muhammad Ahtisham Aslam** [1]

1. Department of Information Systems, Faculty of Computing and Information Technology, King Abdulaziz University, P.O. Box 42808, Jeddah 21551, Saudi Arabia; malmalki0408@stu.kau.edu.sa (M.D.A.); daljeaid@kau.edu.sa (D.A.); maaslam@kau.edu.sa (M.A.A.)
2. Effat College of Business, Effat University, P.O. Box 34689, Jeddah 21478, Saudi Arabia; avisvizi@gmail.com
* Correspondence: ksaeedi@kau.edu.sa; Tel.: +966-126-952-000 (ext. 26157)

**Abstract:** An emerging technology with a secure and a decentralized nature, blockchain has the potential to transform conventional practices in an efficient and dynamic manner. However, migrating to blockchain can be challenging due to the complexity of its infrastructure and processes. The complexity of building applications on blockchain has been highlighted by many studies, thus stressing the need to investigate practical solutions further. A commonly known software engineering concept, software design pattern contributes to the acceleration of software development. It offers a holistic reusable solution for commonly occurring problems in a given context. It helps to identify problems that occur repetitively and describes best practices to address them. The present study is one of the first investigations to inquire into design patterns for blockchain application. Seeking to reduce the complexity in understanding and building applications on blockchain, this paper identifies a design pattern elicitation framework from similar blockchain applications. Next, it provides a demonstration of the Proof of Integrity (PoI) pattern elicited from two different applications on the blockchain. The applicability of the pattern is evaluated by building a blockchain application to verify the integrity of the academic certificates and by explaining how this integrity has been achieved empirically.

**Keywords:** blockchain; software design pattern; integrity; authentication; Proof of Integrity

## 1. Introduction

Blockchain is a distributed ledger that contains a set of sequenced blocks of data [1,2]. Each block records transaction data in a transparent, immutable, and secure fashion [2,3]. This offers a secure platform empowered by furnishing integrity, efficiency, and productivity, time and cost reduction for businesses. However, this platform is complex [4–6]. The literature outlines the complexity of blockchain from technical and business perspectives. The technical perspective has been widely addressed, focusing on technical infrastructure and processes such as peer-to-peer network [7,8], distributed data structure [2,8], database [3,4,9], secure blocks of data [1], consensus protocols [10], and encryption techniques [11]. However, there is limited research pertaining to the business perspective, which is focused on adopting blockchain in business transformation, transactions between participants, and impact analysis [5,12,13].

In this research, we focus on a holistic view of blockchain application. A holistic view addresses what a system analysis requires to design a new solution on blockchain. This is important for supporting the transformation into blockchain. It is important to facilitate understanding of blockchain from both business and technical perspectives and then to develop the confidence to migrate to blockchain.

A lack of research that caters to a holistic view of blockchain implementation is also identifiable in literature [9].

Adopting the design pattern concept from software engineering, this paper proposes that design pattern is a viable solution for addressing the issue focused upon in this study. This is because it captures the similarities between software in a generic and abstract fashion. The pattern offers a standard solution to problems occurring again and again in different domains. Using design pattern fosters understanding of the problem and facilitates designing of the software solution. Moreover, creating a pattern helps to share knowledge for rapidly developing solutions without sacrificing the quality and service expected from solutions [3]. This research argues that adopting design pattern in blockchain application development can enhance understandability and usability of the blockchain system. This argument is aligned to a generic set of patterns for blockchain application [9]. However, our work provides further depth by exploring the pattern technical design concept with reference to empirical cases.

The paper introduces a pattern elicitation framework for blockchain application that is based on capturing the commonalities among different applications through a sequence of three steps, identification, analysis, and abstraction, directed at identifying a common functionality among applications with a common blockchain setting. Analysis of the blockchain application structure using design pattern language [14,15] within each application demonstrates an abstract and generic pattern structure using the design pattern language. The paper demonstrates the process of adopting the framework through empirical elicitation of the Proof of Integrity (PoI) pattern from two blockchain applications, namely, land ownership and intellectual property, and then evaluates the applicability of the identified pattern in a third application called e-certificate. The e-certificate application is developed seamlessly on top of the blockchain network with the design guided by the identified PoI pattern.

This paper starts with a background section on blockchain, highlighting blockchain application architecture, characteristics, and challenges. This section demonstrates the complex nature of the blockchain platform and the utility of adopting concepts from conventional application development. In Section 3 the design pattern concept is highlighted along with the value of adopting design pattern for blockchain applications. Section 4 depicts the pattern elicitation framework for the blockchain application proposed in this paper. The framework steps are illustrated with two examples in order to give an example of adopting the framework. Section 5 provides an empirical evaluation of the framework by adopting a third framework case. Finally, the paper ends with a brief summary of the work conducted in this research

## 2. Blockchain's Application Architecture, Characteristics, and Challenges

The power of blockchain lies in its secure and distributed architecture. It offers a secure decentralized platform for peer-to-peer transactions without the need for a trusted central authority. Blockchain complements the current technologies to integrate conventional business practices with a new level of efficiency and security. However, building an application on the blockchain platform is a complex process, compared to making use of the conventional distributed system. This complexity increases due to the technology stack features of the blockchain infrastructure. This section highlights the main features of blockchain infrastructure and characteristics that are important to understanding the prior building of a blockchain application. The challenges of adopting blockchain applications are also highlighted as these propel the present research to the goal of defining the design pattern for blockchain applications.

In a nutshell, a blockchain is composed of a set of blocks, with each block containing a header and transaction list [10]. The block header is made up of a root and a hash that uniquely identifies the previous block. The transaction list contains sender, transaction, and receiver data, which are secured by an encryption code [7]. Most of the studies have described blockchain architecture as having a number of layers [2,3,16,17]. These layers are broadly highlighted in a generic layered architecture, including application layer, architecture layer, and physical layer, as shown in Figure 1.

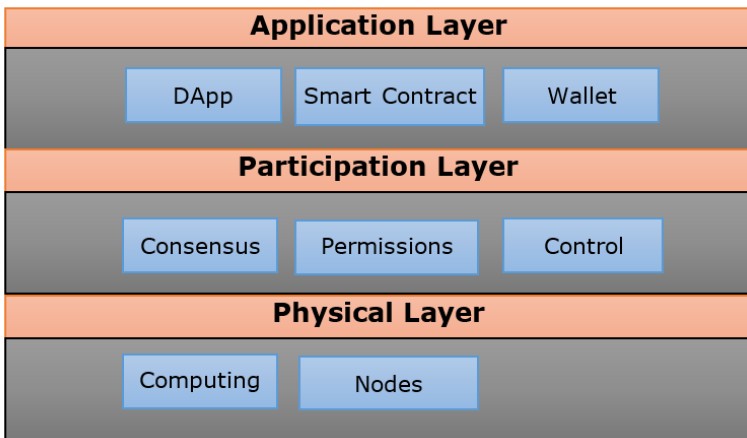

**Figure 1.** Blockchain application architecture.

The application layer reflects the business domain of the blockchain application, the gateway for users to access and use the blockchain. As well as this, it is responsible for the logic and operation of transactions. It comprises the decentralized application (DApp), smart contract, and wallet. DApp stands for decentralized application, and it is the front end application run on top of blockchain technology. A smart contract is a computer code logic executed by a node within the network, and its results are recorded in the blockchain [18]. It manages a set of rules and procedures for transactions in the blockchain. It is worth mentioning that the DApp uses application program interfaces (APIs) to communicate with the blockchain through smart contracts. The wallet, also known as the digital wallet, takes care of authentication and ownership of transactions or assets on the blockchain application through cartography, i.e., through storing both public and/or private keys. This is the first of the aspects to secure the blockchain.

The participation layer is responsible for setting the layout features of the blockchain. It includes different types of aspects that influence the design decision and scope of participation with the blockchain. These layers include consensus, permissions, and control. The consensus is an algorithm used for the decision-making process among nodes in the network, which makes it possible to accept any transaction in the blockchain. It is another security aspect in the blockchain that insures the transaction validity and history. There are a number of consensus algorithms, which can be of three types, including a computation-based, a communication-based, and a hybrid type. The computation-based consensus algorithm is based on solving a complex mathematical puzzle to mine the new blocks such as Proof of Work (PoW). The communication-based type is used when the nodes of the networks accept the new nodes. This type of consensus is mainly used because of a low overhead in computing power and time. Examples of this type of algorithm are Practical Byzantine Fault Tolerance (PBFT), Proof of Stake (PoS), and Proof of Importance. Hybrid types are Proof-of-Elapsed-Time (PoET) and Proof of Authority (PoA) [10].

Permissions in the blockchain identify who can participate in the blockchain, and can be of two main types: permissioned and permission-less [19]. Permissioned means only authorized participants can read. Permission-less means everyone can read the data [20]. These are important to ensure the privacy of data and transactions as well as to determine regulations such as Know Your Customer (KYC) and Anti-Money-Laundering (AML).

Controls validating the transactions can be private, consortial, or public. In the private type, the control is centralized where only one node controls and validates the data, whereas the other nodes only have permission to read. In the consortial type, the control is shared among a few nodes in the network and the rest of the nodes only have read permission. In the public type, such as Bitcoin, all the nodes have permission to write and find consensus [21].

The physical layer refers to the distributed technologies for the blockchain platform, including computing, nodes in the network, and blocks storing the transactions. Nodes in the blockchain

represented as a set of servers are connected and all of them have the same authority. There are two types of nodes, namely the full node and the light node. The full node contains a full copy of the transactions and blocks, and it can be represented as a server. This type of node is responsible for adding light nodes to the network and verifying transactions. On the other hand, the light node contains only the client transactions. A light node is responsible only for transactions used by the client, such as in the case of a student who has a blockchain of records for all of his or her qualifications. The computing component includes the set of computers and storage connected over the network. Each computer works as a server and is represented by a node on a peer-to-peer network.

The computing refers to the computing resources that host the blockchain, such as a high security business network, cloud, and data center. The block is the place where the transactions data are stored. The data are stored in the blocks in hashed form. Hashing is the process of chaining the blocks in the blockchain, i.e., linking a new block to the previous one. Hashing is a computational algorithm that uses the transaction data and address from the previous block as input to produce a fixed size output called a hash [20]. As a hash is irreversible, this gives the blockchain a considerable third layer of security. Storing transaction data in blockchain can be done on-chain or off-chain. The concept of on-chain data refers to all transaction data stored on the blocks. Off-chain data refer to data stored outside the blockchain, such as in cloud storage or in a relational database [8]. The decision of where to store the transaction data is related to the importance of the data and the performance level.

The architecture of the blockchain application presents distinctive characteristics of blockchain applications, which are highlighted as below:

Immutability means that it is impossible to remove or modify any transaction registered on the blockchain. This feature can help to protect the transactions from deletion and counterfeiting [12]. Once a transaction is registered to the blockchain, it will be maintained permanently. The process of saving records in the blockchain is different from that of the traditional database. In the blockchain, we can only add new data and read it, but the records cannot be edited or deleted once recorded. Thus the immutability characteristic makes blockchain a secure and accurate solution for recording transactions and proves that the original data has not been changed [12].

Transparency refers to the viewing access of transactions available to participants who are permitted to access the network. This information includes details of what the transaction is about, the point at which the transaction was added to the blockchain, and users participating in the transaction. The transparency feature is essential for auditing. Blockchain creates a shared technical infrastructure between organizations to allow the participants to track and share the transactions in a transparent and secure fashion [9].

Decentralization means that a full copy of the data is stored on each server (node) on the network. All the nodes have the same features. This depends on the concept of peer-to-peer network [8]. Decentralization can be described as a process where a record of contracts or transactions is stored at several locations accessible to participants without a need for the central authority. In the decentralization environment, all the nodes in the network are informed when a transaction is added. Also, all nodes have a full copy of the transactions [3].

The advantages of an application with these characteristics are promising. Although blockchain has been available since 2008, it is still considered a new technology as it has not been widely adopted by many industries. This has been highlighted by Huaiqing et al. [22], who observed that the adoption of blockchain technology in business is not at an appropriate level of maturity. The reasons highlighted in the literature can be categorized in terms of business and technical challenges. Recent research states that most of the research on blockchain technology tends to focus only on technical perspectives [11] such as security, cryptography, privacy issues, storage, or integration with a legacy system [20].

On the other hand, the business challenges tend to pertain to difficulties of utilization and understanding as well as usability and an unclear impact on business. A number of studies have highlighted difficulties in using and understanding how blockchain works [3,4,6]. This also includes user experience in terms of the difficulty of understanding the technical concepts of blockchain, such as

consensus algorithm, private and public key, wallet, encryption, and hashing. Another challenge is the limited usability of a blockchain application in terms of developer support and end user support [6]. Additionally, recourse limitation also explains why the adoption of blockchain in the wider business domain has been limited [11].

There is dearth of research that addresses the holistic view of blockchain implementation [8,10]. This holistic view is important to facilities in understanding blockchain from both business and technical perspectives and then developing the confidence to migrate to blockchain. As seen in conventional distributed systems, the design pattern, model-driven architecture, and other reusable approaches have a positive impact on the development of blockchain. There is a need for a holistic means to facilitate and assist blockchain implementation as evident in conventional distributed systems. Therefore, this research adopted an integrated design pattern concept within blockchain in order to facilitate the understanding and adaptation of blockchain systems, as design pattern is considered one of the key ways to understand new concepts [14] such as blockchain.

## 3. Design Pattern and Blockchain

This research proposes a holistic approach to improve the usability in building blockchain applications using design pattern. The design pattern is one of the ways to understand new concepts in e-business [15]. Design pattern comprises representation of the solution of a problem in a way that is easily understood by both business users and technical users. It offers a holistic view by providing a description of the architecture, functions, components, and interactions [14].

The patterns for e-business help to clarify the business problem by breaking down the steps of the solution into smaller functions so that the solution can be implemented by using the pattern [23]. Used to establish the primary constructs of the solution, the pattern in software engineering is a useful tool for gaining a better understanding of the relationship between business and technical solutions [23]. The process of design pattern in software engineering helps us to understand a problem occurring commonly within a given context and describes the solution to the problem in such a way that the solution can be used multiple times.

With regard to blockchain, studies have been carried out that discuss creating patterns for blockchain [8,24]. Such research has covered general patterns of blockchain application including Proof of Existence, Proof of Time, Proof of Identity, Proof of Order, Proof of Authorship, and Proof of Ownership. However, the patterns discussed are generic in characterization and unaccompanied by discussion of technical details.

In addition to a design pattern for smart contracts proposed by [25], we found that there was classification of the patterns into Creational Pattern, Structural Pattern, and Inter-Behavioral Pattern. A creational pattern focuses on explaining the creation of smart contracts, whereas the structural pattern helps to manage the relationship among contracts, and the inter-behavioral pattern enhances the flexibility when contract instances operate with each other. All of these patterns focus on the structural design of the smart contract. Their contribution is focused only on that part of smart contracts. These contracts explain patterns from the technical perspective by using a Solidity programming-language-based smart contract. The present research is limited to smart contracts, and other blockchain concepts are not included. Furthermore, the study supports the pattern with examples of how to utilize these patterns within specific business scenarios.

A study proposed by [26] discussed the role of blockchain to solve problems of health care records. They provided an excellent step for addressing the gap and trying to solve the health care interoperability issues. Moreover, they presented a case study and DApp for Smart Health to explain the challenges of duplicated resources and lack of scalability, and how software patterns contributed to solving these problems. However, this study focused on creating patterns in the health care area and was not abstract enough to cover other domains. Also, their study lacked exploration of other potential patterns for handling issues in health care such as security and privacy.

Research on design patterns for blockchain applications is increasing. Some researches focus on the smart contract and data management in general [27], others [26,28–30] focus on platform-specific smart contracts such as Solidity-based smart contract patterns. Fifteen system-level patterns outlined with textual annotation of pattern features supported by real-world examples are proposed in [9].

Limited research exists on the utilizing of patterns to support the usability of blockchain applications, thus creating support for our proposal that design patterns are important for addressing the usability of blockchain applications. Related studies are still limited, and there is a lack of detail in the description of a holistic approach to blockchain application with the support of examples illustrating the utility of identified patterns.

## 4. Pattern Elicitation Framework for Blockchain Application

This section outlines the pattern elicitation framework for blockchain application. We learned from software engineering literature that a pattern can be defined as a solution to a problem that occurs many times. Thus, design patterns are not just concepts, but practical and actionable constructs derived from experience in building real-life applications. Patterns are not created from scratch but are identified and extracted from knowledge in a particular field [14,15]. This mobilized us to design the pattern elicitation framework mainly by capturing the commonalities among similar cases. The framework consisted of three main steps, which needed to be conducted in sequence to identify a pattern, as shown in Figure 2. A detailed explanation of these steps with an empirical example illustrating the Proof of Integrity (PoI) pattern is presented below:

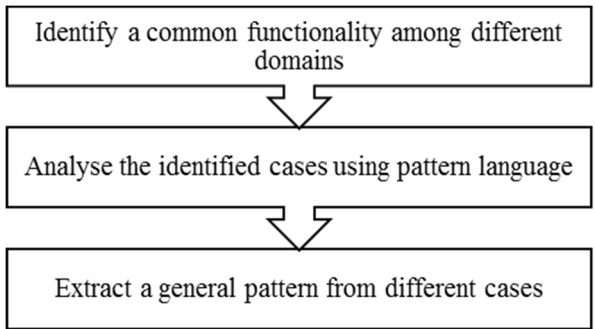

**Figure 2.** Pattern elicitation framework for blockchain application.

First Step: Identify a common functionality among different domains. There is a vast range of blockchain applications available in the literature. To identify common sets of applications with common features, we set specific selection criteria. Our selection was based on functional objective, blockchain scope, participant permission, and core asset of transaction. First, it was important to focus on the main functional objective of the pattern needed in the illustration. In this example, a Proof of Integrity for a high-value asset highlighted as immutability is one of the main characteristics of blockchain. The scope is either private or public blockchain. The permission is related to network participants having write or view permission. The core asset of a transaction refers to a digital or physical asset. These are the three features of influence on the architecture decisions of blockchain application.

For a PoI pattern, we scoped our case on private blockchain, whereby privilege is given to participants who can participate in the business network. With regard to permission, we looked into applications that limited the write permission. Lastly, we looked into the blockchain application that captured transaction of a physical asset. In particular, we focused on applications that captured transactions related to a high-value asset. We identified commonalities in two blockchain applications addressing the integrity of land ownership and intellectual property.

Second Step: Analysis of the identified cases using pattern language was undertaken. The two applications were analyzed to demonstrate how integrity was addressed. We used design pattern language to analyze and document the cases [15]. This helped us to understand the solution and capture

the pattern generic features. The design pattern language is used to document patterns by describing solution objectives, participants, and interactions between participants and data. This design pattern language is designed to provide a convenient approach for understanding and explaining solutions. It supports developing software architectures that are at the core of the IBM Rational Software Architect and also supports the pattern-based engineering tool for model-driven development (MDD).

Furthermore, the design pattern language supports pattern-based software development and offers many benefits as follows. First, it reduces the complexity of software designing by breaking down the large system into a smaller one so the problem can be understood better. Second, this framework displays a higher level of abstraction by focusing on the problem and on how the pattern can solve it. Third, following this framework in designing patterns can contribute to improving productivity by reducing the time of development and thus helping developers economize their efforts. Moreover, this can help to increase the reusability of software by encapsulating the best solutions so that they can be used in different cases by various developers. Therefore, we utilized this design pattern language to illustrate the pattern of blockchain application. An illustration of how design pattern language presented the PoI pattern from the two identified applications is discussed as follows:

A. PoI analysis for land ownership application on blockchain

Pattern Name: PoI for land ownership

Pattern Context: The conventional record-keeping systems of land ownership are primitive and susceptible to abuse and fraud [31,32]. There are cases of claiming ownership of properties based on fraudulent recording. These problems lead to mistrust, disturbance, and lack of integrity of the land recording systems.

The current solutions do not address the complete process of ownership and transfer, and considerable manual work is involved. In addition, not only is the error rate higher in the manual process but also settling a real estate transaction is a slow process, typically taking many weeks. Normally, the real estate transaction process involves third-party enforcement of the contracts. The third party might be not trusted enough, or might lack adequate knowledge of ownership dispute. Furthermore, solving these problems with a centralized database can entail a high security risk [31,32].

Blockchain Solution: Blockchain can prove the integrity of the property ownership agreement by grouping all parties on one platform. It allows participants to transact securely without a centrally trusted intermediary like a traditional real estate agent. The entire sequence of transactions is recorded on the blockchain. All the transactions are visible to business parties, and if fraud transpires, it can be detected easily. Integrity is ensured and checked through the hash algorithm technique in blockchain [31,32].

Pattern Components:

Users:

- Issuer [government, real estate agent]
- Buyer [recipient]
- Verifier [buyer, a party demanding contract integrity]
- Seller [landowner]

Systems: Set of blockchain nodes including buyer, seller, buyer's bank, and real estate agent.

Data: The hashes of the agreements that are stored in the blockchain.

Activities to register and verify the land ownership agreement on blockchain:

(1) A seller creates an account using the wallet through a real estate blockchain API (or DApp) and then posts a request to sell the plot of land. This includes registering information about the land and the price.

(2) Interested buyer creates an account using the wallet through the real estate DApp and then registers personal and bank information.

(3) The buyer checks for lands available for selling and selects the appropriate land.

(4) The smart contract in the real estate DApp checks the credit status of the buyer by sending a request to the buyer's bank to check that the buyer can pay for the purchase.

(5) The buyer's bank sends the approval or rejection based on the buyer's credit information to the smart contract.

(6) The smart contract transfers the ownership of the land to the new buyer and creates an agreement between the buyer, the seller, and the buyer's bank. The agreement contains information about the purchase price, purchase date, buyer, and seller.

(7) The blockchain digitally signs the transaction using the hash algorithm and stores the hash on blockchain blocks.

Activities to verify integrity of land ownership registration on blockchain:

(1) To prove the integrity of the agreement, the verifier uploads the electronic ownership agreement in blockchain.

(2) The blockchain generates the hash for the agreement and compares the resulting hash with the hashes stored in the blockchain.

(3) If the resulting hash matches one of the hashes stored on the blockchain, then the agreement is considered authentic, and the integrity and the ownership of the agreement have been proved.

(4) Figure 3 describes the activities for Proof of Integrity to verify land ownership.

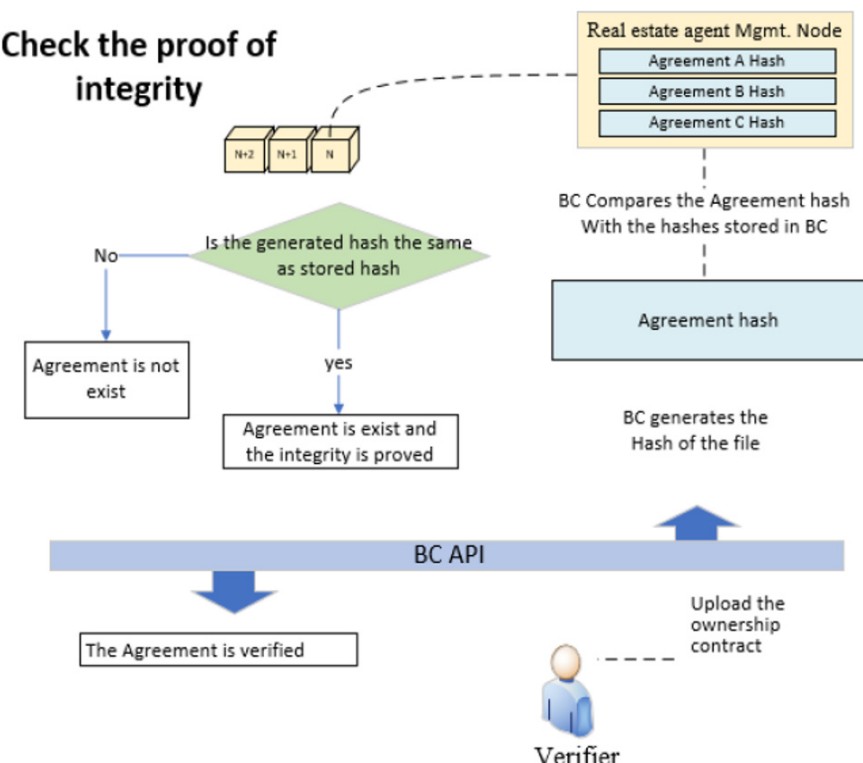

**Figure 3.** PoI structure for land ownership.

Benefits of using blockchain in land registration: The use of blockchain can solve many issues, which include the incidence of fraud in the process of title transfers, the absence of the electronic recording, problems of slowdown in registering real estate transactions that typically take a long time to settle. Blockchain provides accurate records for high-value property such as real estate, and it can identify the current owner and provide proof of the intended owner. All involved parties have authentic digital copies of the agreements. Furthermore, blockchain secures all the transactions using the hash to prove ownership and identity and the hash algorithm to prove existence and authentication [31,33].

B. PoI for intellectual property application on blockchain

Pattern Name: PoI for intellectual property

Pattern Context: The industry of intellectual property (IP) is an essential domain as new ideas, patents, and brands are becoming prevalent. In the current scenario, each country around the world has its own local patent office and a special system to register and check IP. However, the process of checking the uniqueness of the patents in many offices around the world is difficult and requires more time, cost, and efforts.

Meanwhile, as a result of the global nature of the IP industry, the process of checking the authentication for a patent should be carried out at the international marketplace level so as to avoid duplication of patents. The current practices are complicated by paperwork and complexity across the globe, which is costly and time-consuming [5,12]. Proving integrity and authorship of intellectual property and patents in the international marketplace is a challenge due to the fact that information does not exist in a single source. Furthermore, the users sometimes do not know where to look for current inventions and where to protect their new ideas.

Blockchain Solution: Blockchain solves the problem by grouping all the parties in one network, which includes IP regulators, inventors, and invention records. Blockchain helps to keep a secure record of intellectual property on one platform, which can help to simplify the process of verifying the integrity of the patent record. Also, using blockchain to register patents and intellectual property (IP) can reduce the overall number of contract disputes [34–37].

Pattern Components:

Users:

- Issuer [patent office]
- Recipient [inventor]
- Verifier [a party demanding patent integrity]

Systems: The set of nodes that represent patent offices, and nodes of inventors. Blockchain network/platform, blockchain API.

Data: Set of hashes that represent records of the patent.

Activities to create an invention record on blockchain:

(1) The inventor establishes an account on the blockchain network. The account contains a private and public key (address).
(2) The inventor registers information of his or her work through an API blockchain by creating a new transaction, inputs all the required information such as personal information and invention information, and then submits it.
(3) The IP regulators check the patent and then the information of the transaction registered at patent office node.
(4) Blockchain creates the agreement of intellectual property and signs it by using the digital signature. Hash involves using SHA256 algorithm and encrypting in accordance with the inventor's private key.

Activities to verify the integrity of the invention record on blockchain:

(1) The verifier can verify the integrity of an invention by uploading the agreement to the blockchain network and blockchain can provide the hash value for this agreement.
(2) The blockchain compares the hash value with the hashes stored in the blockchain.
(3) If the resulting hash matches one of the hashes stored on the blockchain, the agreement is authentic, and blockchain proves the existence, integrity, and authorship of the invention [35–37].

Figure 4 describes the activities for Proof of Integrity that verify intellectual property.

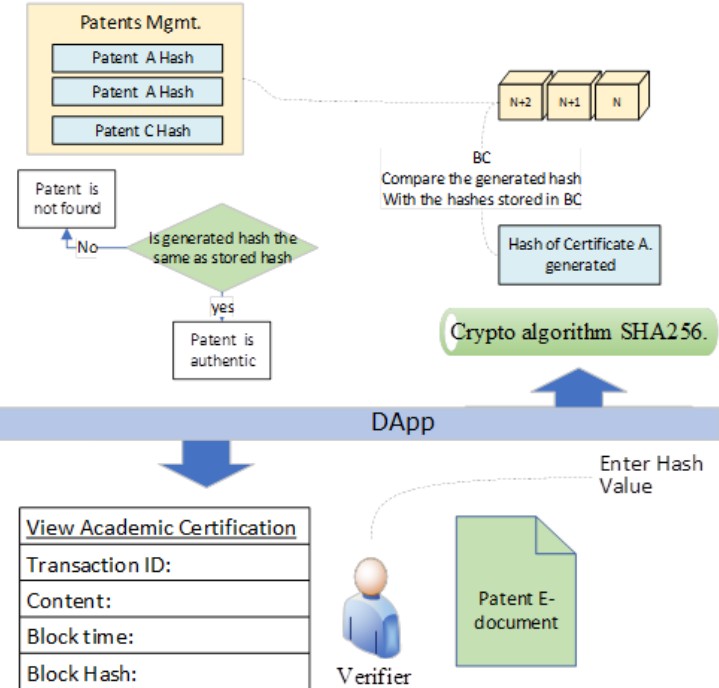

**Figure 4.** PoI structure for intellectual property.

Benefits of using blockchain in intellectual property:

(1)    Blockchain can be used as an authentic digital store for intellectual property.
(2)    The notarizing work will be reduced or at least better adapted to online processes.
(3)    Blockchain can replace intermediaries, thus saving money.
(4)    Blockchain can provide electronic data as legal evidence.
(5)    Blockchain reduces the potential for litigation due to trustworthy proof of ownership for digital assets.
(6)    Blockchain can convert the paper audit trails to digital records while keeping the data immutable.

Third Step: Extract a generic pattern from different cases. After analyzing the application using pattern language, this step captures the commonalities among the applications in abstraction from their domain. Generalization and abstraction are two features that foster pattern usability. The rest of this section outlines the PoI pattern captured from the two applications. The pattern is represented below by explaining how it achieves integrity using blockchain. The representation of the PoI pattern is also demonstrated through the standard design pattern language cases [16] used in the Second Step.

Pattern Name: Proof of Integrity

Pattern Context: The Proof of Integrity (PoI) pattern can be used in applications that issue records related to physical assets and verify the authenticity of the issued record. The record is the data related to physical assets such as certificates, documents, data records, agreements, or contracts. The integrity of the record is an important issue to address, especially in a distributed environment where different parties are involved. Addressing record integrity using blockchain takes care of problems such as counterfeiting and fraud in claiming ownership, authorship, authenticity, and identity.

Blockchain Solution: The PoI pattern for blockchain application addresses integrity proof in a distributed environment. The pattern offers a solution for applications run over private blockchain. The blockchain is set with permission to write for record issuers and permission to view for record verifiers. The record captures the date of when a physical asset is hashed and stored in the blockchain block. The process of integrity check is performed through DApp, which communicates with the blockchain.

Pattern Components:
Users:

- Issuer [record creator]
- Recipient [record owner]
- Verifier [a party demanding record verification]

Systems: The set of nodes in the blockchain network and a private blockchain platform, DApp, and blockchain network.

Data: A set of hashes that represent records of a high-value asset are saved on blockchain blocks.

Proof of Integrity Pattern Processes:

Process 1: Creating a record in blockchain:

(1) Issuer creates a record of an asset in e-document format, such as an agreement, contract, and certificate.
(2) A hash is created for the record using hash function.
(3) The hash is stored in the blockchain's block.

Process 2: Verifying a record in blockchain:

(1) Through DApp, the verifier uploads the e-document into the blockchain.
(2) A hash code is generated for the e-document using the blockchain hash function.
(3) A check of the generated hash and the stored hashes in the blockchain is carried out.
(4) The e-document is authentic if the hash is found in the blockchain, and if it is not found then the e-document is not authentic.

Figure 5 illustrates the verifying process of the PoI pattern. The e-certification application is built by using multi-chain and testing the process of verifying certificates [38].

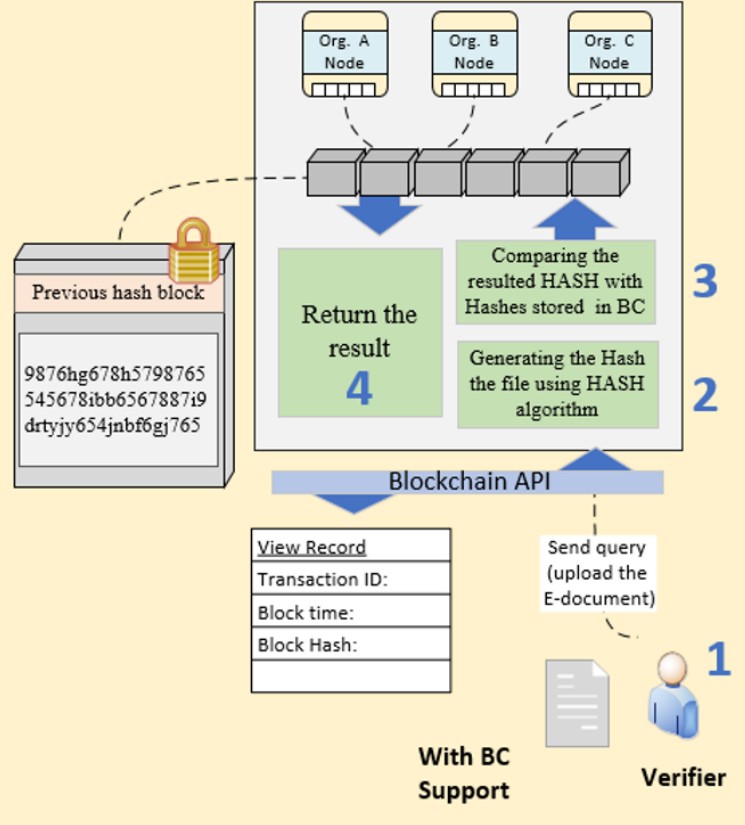

**Figure 5.** Verifying process for the PoI pattern.

PoI Pattern application:

The PoI pattern can be used to store and verify the authenticity of e-documents related to physical assets, including certificates, contracts, agreements, and patents. It can be used as proof of ownership, authorship, authenticity, and identity.

PoI Pattern benefits:

(1) Protects the integrity of records in a decentralized platform.
(2) Increases availability through use of one platform for inquiring into the integrity of records.
(3) Improves service response rate and reliability.
(4) Increases trust and augments transparency.
(5) Generic pattern can capture records of assets from different domains, such as government, real state, health care, and education.

## 5. Empirical Evaluation

The paper presented a pattern elicitation framework with empirical demonstration of a PoI pattern captured from two blockchain systems. This section evaluates the applicability of the PoI pattern in a third application for e-certificates. To conclude, the applicability of the identified pattern in three applications was tested. Although we built the e-certificate system from scratch, using the PoI pattern helped us to design and build the application on blockchain with certainty. The pattern demonstration of the problem and the solution component were used as guidelines to build the application. A demonstration of the adoption of PoI for the e-certificate application structure is presented as below:

Pattern Name: Proof of Integrity for e-certificates on blockchain

Pattern Context:

Students receive their certificates upon graduation. They use these for employment or further education. Some universities or employers ask for verification of certificates, which normally takes up to three weeks. Using a blockchain to verify certificates provides responsiveness, transparency, and simplicity in the authentication process.

Blockchain Solution:

Blockchain can provide a secure platform to store certificates. Educational institutes issue digital certificate to their graduates. This digital certificate is passed to the future employer or educational institutes to use for certificate verification.

Pattern Components:

Users:

- Issuer [academic institutes]
- Recipient [graduates]
- Verifier [employer or academic institutes]

Systems: A set of nodes in the blockchain network, private blockchain platform, DApp, and blockchain network.

Data: Set of hashes that represent e-certificates are saved on blockchain.

Proof of Integrity Pattern for e-certificate processes:

Evaluation of the integrity of e-certificate is based on checking the hash value of the digital certificate by comparing it to the hash stored on the blockchain, as shown in Figure 6.

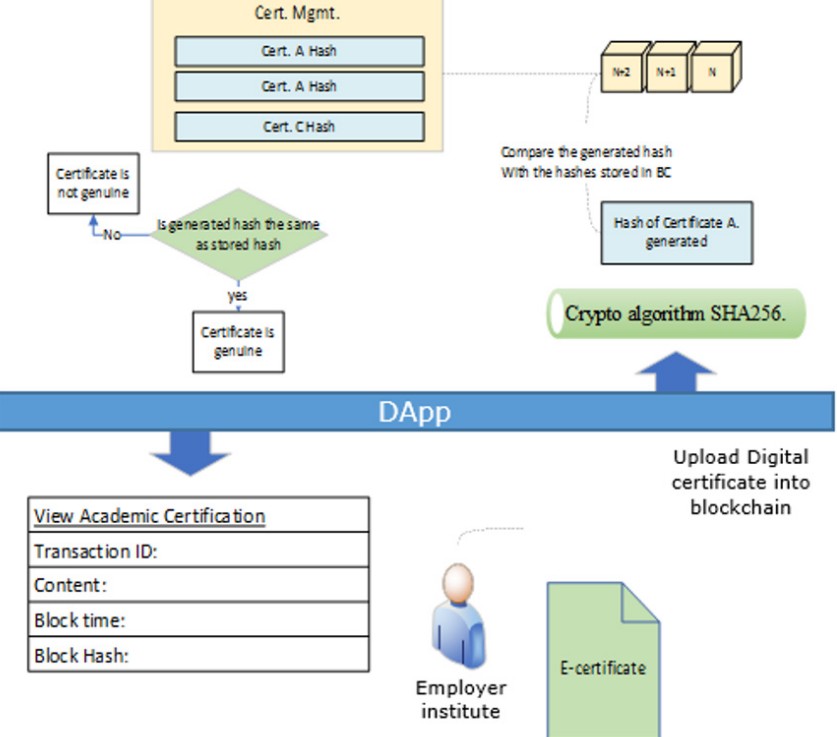

**Figure 6.** PoI pattern for e-certificates.

Process 1: Creating a certificate on blockchain:

(1)　Academic institutes issue e-certificates for their graduates and upload the digital files onto the blockchain.

(2)　A hash value is calculated for each certificate and stored on the blockchain.

Process 2: Integrity proof of e-certificate on blockchain:

(1)　Through DApp, the employer or new university uploads the certificate onto the blockchain.

(2)　A hash code is generated for the e-certificates by using the blockchain hash function.

The value is compared with the hash stored on the blockchain. If the hashes match, the e-certificate is considered authentic

E-Certificate implementation: The implementation took place by making use of an open source platform called Multichain, which hosted the network on an Amazon Web Services (AWS) server. A web interface was created for users to test the platform and this linked the blockchain with an API, as shown in Figure 7. In addition, it generated the hash for the uploaded certificate. The two main functions of the PoI pattern that interacted with the blockchain e-certificate creation and e-certificate verification offer further details on the implementation in [38]. A snapshot of the code for registering the certificate in order to create the certificate on the blockchain is shown in Figure 8. The function was aimed at verifying the integrity of the certificate, as shown in Figure 9.

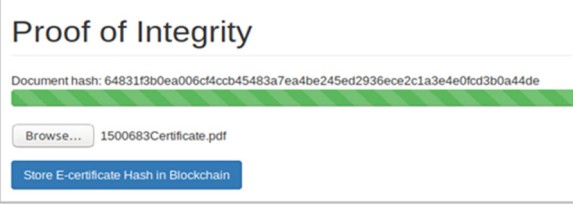

**Figure 7.** Web page generating and uploading hash.

```
// register and publish the hash over BC platform
$app->post('/publish/{signature}', function (Request $request,
Response $response) {

$signature = $request->getAttribute('signature');
    $client = new MultichainClient("http://54.245.198.46:4386",
'multichainrpc', 'EPBGafcRMDXdJtRino4eZgwtzPtB4biPPxvuSfQ7VHin',3)
```

**Figure 8.** Registration of certificate's hash on the blockchain.

```
//Part of verify
$app->get('/verify/{signature}', function (Request $request,
Response $response) {
    $signature = $request->getAttribute('signature');
    $client = new MultichainClient("http://54.245.198.46:4386",
'multichainrpc', 'EPBGafcRMDXdJtRino4eZgwtzPtB4biPPxvuSfQ7VHin',3)
    $data = $client->setDebug(true)-
>executeApi('liststreamkeyitems', array("poe", $signature));
```

**Figure 9.** Code to retrieve the certificate's hash for certificate verification.

## 6. Conclusions

Blockchain is an evolving technology that guarantees a secure platform for distributed applications. However, this new technology is complex. This complexity is attributable to the new application blockchain architecture, which is difficult to comprehend. In another words, identifying a business case that can successfully be implemented in blockchain and also identifying the design feature of the blockchain application are difficult. In a conventional environment, there are a range of software engineering approaches used to address this complexity, such as pattern, web services, and components. These solution help to reuse a solution design for different contexts. Design pattern language is a commonly used approach in software engineering where a pattern is reused to aid in understanding a common problem and to provide a proven design for a problem. Using this approach aids in identifying common business cases and best practices for blockchain solutions. Therefore, the usability in the context of this paper, is to reuse the identified pattern in designing a blockchain application.

The paper highlighted a pattern elicitation framework for blockchain application. A novel conceptual framework consisted of three steps to document patterns for blockchain application. The framework was based on capturing the commonalities among applications in a generic and abstract form. Using design pattern language, the paper then analyzed the application components and documented them. The paper also demonstrated the usability of the framework through empirical examples to capture the PoI pattern with two applications in different domains (land ownership and intellectual property). Thus, business cases that require record integrity can use this pattern. This was validated by using the proposed pattern to build a blockchain application for a third domain (e-certificate).

This paper highlighted the adaptation of the design pattern language for a PoI case in blockchain application. The proposed framework can be used to document further patterns for blockchain application such as KYC, insurance claims, etc. Following the steps of this framework will guide software engineers in identifying the business cases and common solutions for blockchain applications.

**Author Contributions:** Conceptualization K.S., M.D.A. and A.V.; methodology, K.S., M.D.A., M.A.A. and D.A.; software M.D.A. and K.S.; validation, M.D.M., D.A., A.V. and K.S.; writing—original draft preparation, K.S., M.D.A. and D.A.; writing—review and editing, K.S., M.A.A. and A.V. All authors have read and agreed to the published version of the manuscript.

**Funding:** This research received no external funding.

**Conflicts of Interest:** The authors declare no conflict of interest.

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
