# Peer review of "Design Pattern Elicitation Framework for Proof of Integrity in Blockchain Applications"

_sustainability, doi:10.3390/su12208404_

Round 1

Reviewer 1 Report

The article adopted a design pattern language approach, a well-known concept in software engineering for enhancing the understandability and usability of blockchain application design. The paper highlighted a pattern elicitation framework for blockchain application. This article is worth to be published in Sustainability. There are still some minor comments:

  1. The review of existing work is less in the article, and it is recommended to add.
  2. At the end of the “Introduction” section clearly illustrate the sections i.e. the subject discussed in each section. e.g. Section 2 discusses Blockchain’s application architecture, characteristics, and challenges. In section 3 the design pattern of blockchain is illustrated and so on.
  3. Abbreviations such as “AML” and “API” need an illustration. Please, thoroughly check all the abbreviations in the manuscript.
  4. Clearly elaborate with an example on which usability issues this blockchain framework will overcome.
  5. On page 9, The resolution of Figure 4 is not good. Text in white blocks is not visible. Figures should be clearly visible, and text should be understandable even when the reader zooms it. 
  6. PoI represents both proof of integrity and proof of importance. This creates confusion for the reviewer. The abbreviations should be unique in a manuscript.
  7. Some words are repeatedly used in the same paragraph. e.g. the word participants. Too much repetition increases the plagiarism with the same text. Hence, revise it. 
  8. Please illustrate the results more clearly.
  9. Two sections i.e. “Tashkeel Evaluation: Technical Term Translation Case” and “Conclusion” is having the same section number. i.e. 5.

Author Response

1- The review of existing work is less in the article, and it is recommended to add.

A new paragraph is added in the third section where 6 resent publication related to design pattern for blockchain are briefly illustrated

2- At the end of the “Introduction” section clearly illustrate the sections i.e. the subject discussed in each section. e.g. Section 2 discusses Blockchain’s application architecture, characteristics, and challenges. In section 3 the design pattern of blockchain is illustrated and so on.

The following paragraph is added at the end of section 1:

This paper start with a background section on blockchain highlighting blockchain application architecture, characteristics and challenges. This section demonstrate the complex nature of blockchain platform and the eager of adopting concepts adopted in conventional application development. In section 3 the design pattern concept is highlighted and the value of adopting design patter for blockchain applications. Section 4 depicts the pattern elicitation framework for blockchain application proposed in this paper. The framework steps are illustrated with two examples in order to give an example of adopting the framework. Section 5 provides an empirical evaluation of the framework by adopting framework third case. Finally, the paper ends with a brief summary of the work conducted in this research

3-Abbreviations such as “AML” and “API” need an illustration. Please, thoroughly check all the abbreviations in the manuscript.

Added => Anti Money Laundering (AML)

Added => Application Program interface (API)

4-Clearly elaborate with an example on which usability issues this blockchain framework will overcome.

The following paragraph is addressed in the conclusion section which has been extended by the definition of usability in this paper and how the three example used to validate the usability of the identified pattern (land ownership and intellectual property and e-certificate).  

See second and third paragraphs of the conclusion section

5-On page 9, The resolution of Figure 4 is not good. Text in white blocks is not visible. Figures should be clearly visible, and text should be understandable even when the reader zooms it

Figure 4 is replaced by bigger Figure

6-PoI represents both proof of integrity and proof of importance. This creates confusion for the reviewer. The abbreviations should be unique in a manuscript.

I just realized that abbreviation is the same for proof of integrity and proof of impotence. The proof of integrity is what we use to evaluate the proposed framework. Whereas, Proof of Importance is one of the available consensus algorithms in blockchain. To avoid confusing I removed the appreciation from, Proof of Importance which is mentioned one time only in the second section

7-Some words are repeatedly used in the same paragraph. e.g. the word participants. Too much repetition increases the plagiarism with the same text. Hence, revise it. 

The word participants in this paper refers to parties who are permitted to access the network. I used it as it is the term commonly used to describe parties participating in blockchain network. I prefer to keep it but if you are not convinced I will change it

8-Please illustrate the results more clearly.

The following paragraph is addressed in the conclusion section which has been extended

See third and fourth paragraphs of the conclusion section

9-Two sections i.e. “Tashkeel Evaluation: Technical Term Translation Case” and “Conclusion” is having the same section number. i.e. 5.

There was a mistake on the section title which now changed to 5. Empirical evaluation

Reviewer 2 Report

The authors present the interesting information about design patterns for blockchain applications and aim to reduce the complexity of building applications on it. The authors claim that it is the first investigation of design patterns for blockchain applications.

Overall, the paper looks good and provides a straightforward (easy to read) material about blockchain applications.
But, it is not clear what is exactly the contributions of this paper. I suggest to authors to outline their contributions.

Some more comments:

Point 1:

In Abstract, rows 4-5 the word “highlighted” is repeated twice.

Point 2

I don’t understand the first sentence in convolution “The paper highlighted the complexity of blockchain as a new platform for distributed applications.”

Point 3

Something wrong with references  eg.  Reference “Huaiqing et al.[14]” at page 4 is missing. Please correct.

Author Response

But, it is not clear what exactly the contributions of this paper is. I suggest to authors to outline their contributions.

The contribution of the paper is providing a pattern elicitation framework for blockchain application adopting a design pattern language approach, a well-known concept in software engineering to enhance usability of application development. It is adopted in web base development such as e-commerce. In this paper the design pattern language approach is adopted in the design of blockchain application.  

Point 1:In Abstract, rows 4-5 the word “highlighted” is repeated twice.

The highlighting is changed using another word (stress)

Point 2: I don’t understand the first sentence in convolution “The paper highlighted the complexity of blockchain as a new platform for distributed applications.

This statement is rephrased with to è Blockchain is an evolving technology, guarantees a secure platform for distributed applications.  However, this new technology is complex

Point 3:Something wrong with references  eg.  Reference “Huaiqing et al.[14]” at page 4 is missing. Please correct.

Reference is updated